# Effects of Co-Culture EBV-miR-BART1-3p on Proliferation and Invasion of Gastric Cancer Cells Based on Exosomes

**DOI:** 10.3390/cancers15102841

**Published:** 2023-05-19

**Authors:** Mengyao Lin, Shun Hu, Tianyi Zhang, Jiezhen Li, Feng Gao, Zhenzhen Zhang, Ke Zheng, Guoping Li, Caihong Ren, Xiangna Chen, Fang Guo, Sheng Zhang

**Affiliations:** 1Department of Pathology, The First Affiliated Hospital, Fujian Medical University, 20 Chazhong Road, Fuzhou 350005, China; 2Department of Pathology, National Regional Medical Center, Binhai Campus of the First Affiliated Hospital, Fujian Medical University, 999 Huashan Road, Fuzhou 350212, China; 3Key Laboratory of Systems Biomedicine (Ministry of Education), Shanghai Center for Systems Biomedicine, Shanghai Jiao Tong University, 800 Dongchuan Road, Shanghai 200240, China

**Keywords:** gastric cancer, exosome, small RNA expression characteristics, EBV-miR-BART1-3p, proliferation, invasion

## Abstract

**Simple Summary:**

EBV (Epstein-Barr virus) miRNA is a signaling molecule between infected and non-infected cells, which plays a biological role through exosome delivery. However, the mechanism of its effect on proliferation, apoptosis and immune response of EBVaGC is poorly understood. The aim of this study was to clarify the miRNA expression profiles of EBV-positive and -negative GC cells and their exosomes, and the mechanism of the effect of exosome-based miR-BART1-3p on the biological behavior of GC cells. We found that miR-BART1-3p can affect the growth of tumor cells through the exosome pathway. Co-culture with exosomes with miR-BART1-3p expression silence can improve the proliferation, healing, migration and invasion of GC cells. miR-BART1-3p may regulate the proliferation of GC cells through potential target genes USP37 and MACC1.

**Abstract:**

Aim: EBV encodes at least 44 miRNAs involved in immune regulation and disease progression. Exosomes can be used as carriers of EBV-miRNA-BART intercellular transmission and affect the biological behavior of cells. We characterized exosomes and established a co-culture experiment of exosomes to explore the mechanism of miR-BART1-3p transmission through the exosome pathway and its influence on tumor cell proliferation and invasion. Materials and methods: Exosomes of EBV-positive and EBV-negative gastric cancer cells were characterized by transmission electron microscopy. NanoSight and Western blotting, and miRNA expression profiles in exosomes were sequenced with high throughput. Exosomes with high or low expression of miR-BART1-3p were co-cultured with AGS cells to study the effects on proliferation, invasion, and migration of gastric cancer cells. The target genes of EBV-miR-BART1-3p were screened and predicted by PITA, miRanda, RNAhybrid, virBase, and DIANA-TarBase v.8 databases, and the expression of the target genes after co-culture was detected by qPCR. Results: The exosomes secreted by EBV-positive and negative gastric cancer cells range in diameter from 30 nm to 150 nm and express the exosomal signature proteins CD9 and CD63. Small RNA sequencing showed that exosomes expressed some human miRNAs, among which hsa-miR-23b-3p, hsa-miR-320a-3p, and hsa-miR-4521 were highly expressed in AGS-exo; hsa-miR-21-5p, hsa-miR-148a-3p, and hsa-miR-7-5p were highly expressed in SNU-719-exo. All EBV miRNAs were expressed in SNU-719 cells and their exosomes, among which EBV-miR-BART1-5p, EBV-miR-BART22, and EBV-miR-BART16 were the highest in SNU-719 cells; EBV-miR-BART1-5p, EBV-miR-BART10-3p, and EBV-miR-BART16 were the highest in SNU-719-exo. After miR-BART1-3p silencing in gastric cancer cells, the proliferation, healing, migration, and invasion of tumor cells were significantly improved. Laser confocal microscopy showed that exosomes could carry miRNA into recipient cells. After co-culture with miR-BART1-3p silenced exosomes, the proliferation, healing, migration, and invasion of gastric cancer cells were significantly improved. The target gene of miR-BART1-3p was FAM168A, MACC1, CPEB3, ANKRD28, and USP37 after screening by a targeted database. CPEB3 was not expressed in all exosome co-cultured cells, while ANKRD28, USP37, MACC1, and FAM168A were all expressed to varying degrees. USP37 and MACC1 were down-regulated after up-regulation of miR-BART1-3p, which may be the key target genes for miR-BART1-3p to regulate the proliferation of gastric cancer cells through exosomes. Conclusions: miR-BART1-3p can affect the growth of tumor cells through the exosome pathway. The proliferation, healing, migration, and invasion of gastric cancer cells were significantly improved after co-culture with exosomes of miR-BART1-3p silenced expression. USP37 and MACC1 may be potential target genes of miR-BART1-3p in regulating cell proliferation.

## 1. Introduction

The incidence and mortality of gastric cancer (GC) rank fifth and fourth in the world, respectively [1,2]. TCGA divides gastric adenocarcinoma into four molecular subtypes: EBV-positive (Epstein–Barr virus (EBV)-associated gastric cancer, EBVaGC), microsatellite instability (MSI), genomically stable (GS), and chromosomal instability (CIN). Of these, EBVaGC accounts for about 10% [3]. EBVaGC were subclassified, based on the pattern of host inflammatory immune responses, into three histologic subtypes: lymphoepithelioma-like carcinoma (LELC), Crohn’s disease-like lymphocytic reaction (CLR), and conventional adenocarcinoma [4].

EBV belongs to the human herpes virus type 4 and can cause infectious mononucleosis, lymphoma, nasopharyngeal cancer, and stomach cancer, among other diseases. EBVaGC belongs to latent infection type I, expressing viral gene products such as EBNA1, EBERs, and BART miRNAs [5]. However, the mechanism of how EBV induces the transformation of infected cells into cancer cells remains unclear.

miRNAs are small non-coding single-stranded RNAs with a length of about 20 to 24 nucleotides, which are involved in post-transcriptional gene expression regulation. Viral miRNAs are signaling molecules between infected and non-infected cells [6,7]. The EBV BHRF1 and BART gene clusters encode at least 44 miRNAs [8]. Tumor cells can sort virus-related disease cytokines, including miRNA, tRNA, mRNA, lncRNA, and proteins, loaded into exosomes and delivered to uninfected cells, affecting cell proliferation, apoptosis, and immune response, thus affecting host cell function [9,10,11,12,13,14]. To date, the expression profiles of EBVaGC-secreted exosomes and the human- and virus-associated miRNAs sorted in the exosomes are poorly understood, and the mechanisms of action have not been detailed. Further study of miRNAs in exosomes can clarify the role of miRNAs in exosomes. We used small RNA sequencing technology to analyze miRNA expression profiles in EBV-positive and -negative GC cells and their exosomes.

At present, there are few studies on the role of EBV miR-BART1-3p in the development of tumors. Shinozaki-Ushiku et al. found that the expression level of miR-BART1-3p was the highest in epithelial tumors [15]. Similarly, the expression level of miR-BART1-3p in central nervous system tumors was significantly higher than that in normal specimens [16]. These results suggest that EBV can change the state of cancer cells and participate in the development of cancer by altering the expression of miR-BART1-3p. miR-BART1-3p was transfected into GC cells, and it was found that miR-BART1-3p could induce G0/G1 arrest and inhibit the growth of GC cells, resulting in decreased expression levels of E2F3 mRNA and encoded protein and affecting cell proliferation and differentiation [17]. We silenced the expression of miR-BART1-3p and observed the effects of high and low expression of miR-BART1-3p on the proliferation, invasion, and migration of GC cells by CCK8 assay, cell scratch assay, and Transwell migration and invasion assay.

Min et al. revealed that EBV-miR-BART1-3p inhibited apoptosis of GC cells and promoted cell migration by targeting DAB2 [18] but did not study whether EBV-miR-BART1-3p could play a regulatory role in tumor cell proliferation through exosomes. We isolated and purified exosomes by ultra-high-speed centrifugal method, constructed exosomes for co-culture, and observed the effects of exosomes with high and low expression of miR-BART1-3p on the proliferation, migration, and invasion ability of GC cells. qPCR was used to detect the expression of miR-BART1-3p and its target genes. The establishment of an exosome co-culture experiment would help clarify the role and mechanism of regulating the proliferation and invasion of GC cells by miR-BART1-3p based on exosome delivery.

## 2. Materials and Methods

### 2.1. Cell Culture and Reagents

SNU-719 (EBV-positive GC cell line), AGS-NC (EBV-negative GC cell line), and AGS-OE (lentivirus-transfected miR-BART1-3p overexpression cell line in our laboratory) were cultured in RPMI-1640 and F12 separately. Then, 5% fetal bovine serum without exosome and penicillin and streptomycin (100 U/mL) were added to the medium prior to use. The supernatant was collected when the cells were growing vigorously (up to about 70% fusion).

### 2.2. EBER In Situ Hybridization

SNU-719 cells and AGS cells that grew vigorously in cell culture dishes were trypsinized and centrifuged to prepare cell wax blocks and 4 um thick slices. EBER in situ hybridization was performed to test negative or positive for EBV.

### 2.3. Preparation of Exosome-Free Fetal Bovine Serum

Exosome-free fetal bovine serum was prepared by the ultra-high-speed centrifugal method: (1) Fresh fetal bovine serum was placed in a 50 mL centrifuge tube and centrifuged at 4 °C at 300× *g* for 10 min. (2) The supernatant was carefully drained and placed into another new 50 mL centrifuge tube. The pellet was discarded and centrifuged at 2000× *g* at 4 °C for 10 min. (3) We placed the supernatant carefully into a new 50 mL centrifuge tube and discarded the pellet. The supernatant was centrifuged at 10,000× *g* at 4 °C for 30 min. (4) After the third centrifugation, the supernatant was transferred to the ultra-high-speed centrifugal tube. The supernatant was centrifuged at 4 °C at 100,000× *g* for 120 min. (5) After the fourth centrifugation, the supernatant was carefully sucked out and the sediment was discarded. After completion of the above steps, the supernatant obtained was the exosome-free fetal bovine serum.

### 2.4. Isolation of Exosomes

Exosomes in the supernatant of cell culture were isolated by differential hypervelocity centrifugation: (1) The collected supernatant was placed in a 50 mL centrifuge tube and centrifuged at 4 °C at 300× *g* for 10 min. (2) After centrifugation, the supernatant was carefully absorbed and placed into a new 50 mL centrifuge tube. The supernatant was centrifuged at 2000× *g* at 4 °C for 10 min, and the precipitation was discarded. (3) After centrifugation, the supernatant was carefully absorbed, placed in a new 50 mL centrifuge tube, centrifuged at 10,000× *g* at 4 °C for 30 min, and the precipitation was discarded. (4) After centrifugation, the supernatant was carefully absorbed and transferred to an ultra-high-speed centrifuge tube for 120 min at 100,000× *g* at 4 °C. (5) After centrifugation, the supernatant was discarded and the pellet was resuspended with 200 μL 1 × PBS solution. The resulting pellet mostly contained exosome particles, which were dissolved by repeated pipette blowing. (6) Finally, the dissolved exosomes were filtered through a 0.22 μm filter to remove bacteria and further purify the exosomes.

### 2.5. Exosome Morphology Observed by Transmission Electron Microscopy

The exosomes of 10 μL were absorbed into a 300-mesh Cu mesh with a diameter of 2 mm, and the excess liquid was drained with filter paper. The mesh was negatively stained with 2% pH 7.0 tungstate solution for 2–3 min. Morphological characteristics of exosomes were observed by transmission electron microscopy (Tecnai G2).

### 2.6. Exosome Size and Concentration by Nanoparticle Tracking Analysis (NTA)

The particle size and concentration of exosomes were detected by a nanoparticle tracer analyzer (NanoSight NS300, NTA) (Malvern, UK) (Laboratory of Guangzhou Marvern Panalytical Company). One μL of isolated and purified exosome suspension was diluted with equal volume PBS solution until no more than 100 particle beams were displayed in the NTA software screen. By tracking the Brownian motion of a single particle, the NTA software calculated the motion rate and finally obtained the particle size distribution and concentration information of the whole exosome system.

### 2.7. Extraction of Cell Protein

Cell protein was extracted: (1) Cell medium was discarded and cultured cells were washed 3 times with 1 × PBS solution, and the remaining PBS was absorbed as much as possible. (2) Cells were lysed with 200 μL RIPA at 4 °C for 10 min. (3) Cell lysates were transferred to Ep tubes and centrifuged at 4 °C at 12,000× *g* for 10 min. (4) The supernatant was transferred to a clean Ep tube and 1 μL was absorbed for protein concentration determination. The rest was added to the protein loading buffer (5xSDS) and thoroughly mixed. (5) The protein sample was boiled in boiling water for 10 min to denature the protein sample completely and then stored at −80 °C.

### 2.8. Extraction of Exosomal Protein

Exosomal protein was extracted: (1) Exosomes were mixed with RIPA and PMSF with appropriate cell lysates and cleaved at room temperature for 10 min. (2) After lysis, the sample was centrifuged at 4 °C at 12,000× *g* for 10 min. (3) The supernatant (total exosomal protein) was transferred to a clean Ep tube, and then 2 μL was sucked out to determine protein concentration and the rest was added to the protein loading buffer and thoroughly mixed. (4) The protein sample was boiled in boiling water for 10 min to denature the protein sample completely and then stored at −80 °C.

### 2.9. Determination of Cellular and Exosomal Protein Concentrations

The protein concentration was determined by the BCA method, and the procedure was carried out according to the kit instructions. The absorbance at 550 nm was measured using a microplate analyzer, and the standard curve was drawn to calculate the concentration of protein samples.

### 2.10. Expression Analysis of Exosomal Protein

Expression of exosomal protein was analyzed as follows: (1) Exosome proteins of 60 μg equal samples were electrophoresed in a 5% dense adhesive at a constant voltage of 90 V. When the sample was in a 12% SDS-PAGE adhesive, the voltage was increased to a constant voltage of 120 V for protein separation. (2) After SDS-PAGE electrophoresis, the electrotransfer film was started at 110 V constant voltage. (3) The membrane was sealed with 3% bovine serum albumin (BSA) solution at room temperature for 2 h. (4) Primary antibodies CD63 (1:1000, Affinity Bioscience, OH, USA) and CD81 (1:1000, Affinity Bioscience, OH, USA) were used as exosome marker detection membranes. (5) After washing, the membrane was incubated with goat anti-mouse IgG labeled with horseradish peroxidase (HRP) (1:5000) at room temperature for 50 min. (6) After full washing, the membrane was observed with ECL chemiluminescent substrate. Image Lab 4.1 analysis software was used for gray analysis.

### 2.11. Extraction of Ribonucleic Acid

Total RNA was extracted from cells and exosomes using a high-purity RNA extraction kit (TRIzol) according to the manufacturer’s protocol. The OD260 and OD280 values and ratios of 1 μL extracted samples were determined using an ultraviolet spectrophotometer to determine the concentration and purity of the extracted samples.

### 2.12. Library Construction and Small RNA Sequencing

The miRNA library was prepared using an RNA sample preparation kit (Epicenter, San Diego, CA, USA) according to the manufacturer’s instructions. As outlined below, small RNAs were attached to 3 and 5 splices for reverse transcription to produce single-stranded cDNA. Then, PCR amplification was performed. The constructed library was tested for quality and yield using the Agilent 2100 bioanalyzer and ABI StepOnePlus real-time fluorescence quantitative PCR system. Four small RNA libraries were constructed from RNA samples extracted from EBV-positive GC cell lines (SNU-719) and EBV-negative GC cell lines (AGS) and their exosomes. The library was sequenced on the Illumina HiSeqTM 2500 platform. Clean reads were obtained after completing preliminary data filtering. Then, the classification annotation of ncRNAs, distribution of small RNA sequence lengths, annotation of genomic repeat regions, annotation of functional elements, common and unique sequence analysis, sample correlation analysis, and analysis of miRNA expression differences between samples were performed to obtain EBV-related miRNA expression profiles.

### 2.13. Expression Profile Analysis of EBV-miRNA

The expression profile of EBV-miRNA was analyzed by miRDeep2 and Mirbase.v21. The differential expression of EBV-miRNA was expressed with a heat map, and the expression of all EBV-miRNA was expressed with a rectangle map between two groups of samples (AGS-Cell vs. AGS-exo, SNU-719 Cell vs. SNU-719-exo).

### 2.14. Construction of SNU-719-Inhibitor Cell Line with Low Expression of EBV-miR-BART1-3p

EBV-miR-BART1-3p micrONTM inhibitor is a miRNA inhibitor, a chemically modified, mature miRNA complementary single chain. Cy3 fluorescent labeling inhibitor was used to observe the transfection efficiency under an inverted fluorescence microscope.

### 2.15. Verification of the Expression Level of miR-BART1-3p in Cell Lines and the Expression of Target Genes in AGS Cells after Co-Culture of Exosomes for 24 h

Total RNA of cells or exosomes was extracted. According to the instructions of the miRNA reverse transcription kit, RNA was reversely transcribed into cDNA, and cDNA was used as a template for PCR amplification. qPCR was performed using a two-step method. The fusion curve was prepared to observe the amplification curve and fusion curve of qPCR, and Ct values were read. Further, 2-Ct represented the change multiple of the expression of target genes in the experimental group compared with that in the control group. ΔΔCt = (ΔCt target gene − ΔCt internal reference gene) experimental group − (ΔCt target gene − ΔCt internal reference gene) control group. Sequence of miRNA and internal reference primers, U6: Forward: CTCGCTTCGGCAGCACA, Reverse: AACGCTTCACGAATTTGCGT; EBV-miRNA-BART1-3p: Forward: CCTCTTGGGCATCATCTTGCT, Reverse: GATAGTCCCTTGGTTGGTGCT.

### 2.16. CCK8 Cell Proliferation Experiment

For this experiment, 10^3^ cells of SNU-719, SNU-719-inhibitor, AGS-OE, and AGS-NC were inoculated in 96-well plates, and 200 μL cell suspension was added to each well. After labeling for 0 h, 12 h, 24 h, 36 h, 48 h, 60 h, and 72 h, each well was filled with 10 μL CCK8 solution, gently mixed, and then placed in the medium for static 30 min. We measured the absorbance of each well at 450 nm.

### 2.17. Cell Scratch Experiment

SNU-719, SNU-719-inhibitor, AGS-OE, and AGS-NC cells of logarithmic growth stage were added into 6-well plates to prepare monolayers. Once the cells were covered in the dish, the pipette tip was scratched perpendicular to the plate. The scratched cells were then washed 3 times with PBS. Fresh medium was added, observed, and photographed at 0 h, 24 h, and 48 h. ImageJ was used to measure the relative distance that cells migrate to the scratched area.

### 2.18. Transwell Migration Assay

SNU-719, SNU-719-inhibitor, AGS-OE, and AGS-NC cells were suspended in a serum-free medium with a concentration of 1 × 10^5^ cells/mL. In 24-well plates, 200 μL cell suspension was added to the upper chamber, 700 μL medium containing 10% serum was added to the lower chamber, and the Transwell was incubated in a 37 °C incubator for 24 h. The liquid in the chamber was sucked clean, and the chamber was moved into a clean well. The chamber was cleaned twice with 1 mL PBS, and 700 μL 0.1% crystal violet solution containing methanol-PBS was added for fixed staining for 20 min. The crystal violet solution was sucked, cleaned twice with 1 mL PBS, and the unmigrated cells in the upper layer were wiped off with a wet cotton swab. The chamber was placed on a clean slide, and four fields were observed under the microscope and photographed. ImageJ calculated and analyzed the number of cells passing through the Transwell chamber.

### 2.19. Transwell Invasion

Matrigel was diluted with serum-free cold cell medium 1:9 in a 24-well plate. Then, 100 μL of the Matrigel was added to the upper chamber of the Transwell and incubated in the chamber at 37 °C for 1 h. SNU-719, SNU-719-inhibitor, AGS-OE, and AGS-NC cells were suspended in a serum-free medium with a concentration of 1 × 10^5^ cells/mL. Next, 200 μL cell suspension was added to the upper chamber, and 700 μL medium containing 10% serum was added to the lower chamber and incubated in a 37 °C incubator for 24 h. The liquid in the chamber was sucked and moved into a clean well. The chamber was cleaned twice with 1 mL PBS, and 700 μL 0.1% crystal violet solution containing methanol-PBS was added for fixed staining for 20 min. The crystal violet solution was sucked and cleaned twice with 1 mL PBS, and the unmigrated cells in the upper layer were wiped off with a wet cotton swab. The chamber was placed on a clean slide, four fields were observed under the microscope and photographed. ImageJ calculated and analyzed the number of cells passing through the Transwell chamber.

### 2.20. Absorption of Exosomes by Living Cells Observed by Laser Confocal Microscopy

PKH67 is a lipophilic dye that binds to lipid membranes stably, and PKH67-labeled exosomes emit green fluorescence. To produce an exosome-dye complex, 50 μL PKH67 dye was co-incubated with 10 µg exosome for 10 min. The stained exosomes were extracted using the SBI ExoQuick-TCTM exosome extraction kit. PKH67-exosome was incubated with AGS cells for 24 h and PBS was used as control. DAPI was observed at 405 nm, and exosome-PKH67 staining was observed at 488 nm.

### 2.21. Expression of miR-BART1-3p after Co-Culture with GC Cells in Exosomes with High Expression or Silence of miR-BART1-3p and Its Effect on Proliferation and Invasion

First, 20 μg exosomes derived from SNU-719, SNU-719-inhibitor, AGS-OE, and AGS-NC cells and PBS as control were added to AGS for 24 h. The expression of miR-BART1-3p in AGS cells after 24 h co-culture with exosomes was detected by qPCR. CCK8 cell proliferation assay, cell scratch assay, Transwell cell migration assay, and invasion assay were used to detect the effects of co-culture with exosomes on the proliferation and invasion ability of AGS cells.

### 2.22. Target Gene Analysis of miR-BART1-3p

miR-BART1-3p was analyzed by miRanda (https://microrna.org/microrna/getMirnaForm.do) (accessed on 24 Dec, 2022), PITA (https://genie.weizmann.ac.il/pubs/mir07/mir07_dyn_data.html) (accessed on 24 December 2022), RNAhybrid (https://bibiserv.techfak.uni-bielefeld.de/) (accessed on 24 December 2022), virBase (https://www.rna-society.org/virbase) (accessed on 24 December 2022), and TarBase (https://www.microrna.gr/tarbase) (accessed on 24 December 2022) database. According to the degree of perfect complementary pairing of the miRNA seed region, the sequence conservatism of MRE (miRNA recognition elements), the binding free energy (ΔGduplex) of miRNA-mRNA duplex, the sequence characteristic analysis of target molecules, and the reference data of miRNA target genes in experimental support, combined with the prediction results, five database intersection target genes were obtained by Venn diagram.

### 2.23. Statistical Analysis

SPSS 22.0 software was used for statistical analysis. The difference in miRNA expression between cells and exosomes was detected by paired sample T-test. |change|log2 (folding) and edge R were used to calculate the miRNA expression differences in the cells and exosomes. The correlation of miRNA expression was calculated using Pearson’s correlation coefficient. A Chi-square test was used to compare gene expression levels, a Mann–Whitney test was used to compare the differences in expression levels between groups, and a *t*-test and ANOVA were used to analyze the overall differences. *p* < 0.05 was considered statistically significant.

## 3. Results

### 3.1. Characterization of miRNA in Exosomes of EBV-Positive GC Cells

#### 3.1.1. Morphology of AGS and SNU-719 GC Cells

AGS cells are fusiform, prismatic, epithelioid, and adherent. SNU-719 cells are small round clumps with semi-adherent growth. EBER in situ hybridization staining showed that AGS was EBVnGC (EBV-negative gastric cancer) cell line and SNU-719 was EBVaGC cell line (Figure 1A–D).

#### 3.1.2. Structure of Exosomes Secreted by SNU-719 and AGS Cells

Exosomes of SNU-719 and AGS cells were obtained by ultrafast differential centrifugation and their morphology was observed by transmission electron microscopy. Figure 1E,F presents these exosomes as membranous vesicles between 30 and 150 nm in diameter, with round translucent cup structures and slightly sunken discoid vesicles, as reported in the literature. There was no significant difference in the morphology of exosomes secreted by SNU-719 and AGS cells.

#### 3.1.3. Size of Exosomes Secreted by SNU-719 and AGS Cells

The results of NTA showed that the extracted exosome particles were evenly distributed with low dispersion. AGS-exo and SNU-719-exo had maximum peaks at 113.4 nm and 121.2 nm, respectively, indicating that the extracted exosomes were within the normal particle size range (30–150 nm) (Figure 1H–I).

#### 3.1.4. Expression Analysis of Exosomal Proteins Secreted by SNU-719 and AGS Cells

Western blotting results showed that AGS-exo and SNU-719-exo detected strong band signals at the molecular weights of 24 kDa and 28 kDa, respectively. CD9 and CD63 were exosome-specific membrane protein molecules (Figure 1G).

#### 3.1.5. Distribution of Small RNAs in Cells and Exosomes

##### Annotation of ncRNAs in SNU-719 and AGS Cells and Exosomes

Figure 2A shows the annotation of ncRNAs in SNU-719 and AGS cells and exosomes. SNU-719 and AGS cells and exosomes contained miRNA, rRNA, tRNA, snRNA, piRNA, and other small RNAs, among which miRNA was the most abundant type of small RNA.

##### Small RNA Sequence Length Distribution in SNU-719 and AGS Cells and Exosomes

Figure 2B shows that the length distribution of small RNA sequences (17–27 nt) from SNU-719 and AGS cells and exosomes was unique. The length of small RNAs in AGS cells and AGS-exo ranged from 17 to 25 nt, with the highest abundance being 23 nt, followed by 22 nt. The length range of small RNA in SNU-719 cells and SNU-719-exo was mainly 17–26 nt, with the highest abundance being 22 nt, followed by 23 nt.

##### sRNA Analysis of Genomic Repeat Regions in SNU-719 and AGS Cells and Exosomes

Figure 2C shows the results of sRNA analysis in the genome repetition region: tRNA was most common in AGS, followed by scRNA and rRNA (long interspersed nuclear element, LINE). tRNA was the most common in AGS-exo, followed by rRNA and scRNA. rRNA was most common in SNU-719, followed by tRNA and scRNA. tRNA was most common in SNU-719-exo, followed by scRNA and rRNA. The results showed that sRNA in the repetitive regions of the genome was basically the same in the four samples and mainly concentrated in the tRNA, rRNA, and scRNA regions.

##### Analysis of Functional Elements in SNU-719 and AGS Cells and Exosomes

As shown in Figure 2D, analysis results of functional elements in SNU-719 and AGS cells were mainly located in the non-coding RNA exonic region (ncRNA_exonic region), and their exosomes were mainly located in the intergenic region (intronic region). The number of exons, ncRNA_intronic, up/, and 3′ UTR or 5′ UTR readings (UTR5/UTR3) was relatively small.

##### Common and Unique Sequence Analysis of SNU-719 and AGS Cells and Exosomes

Figure 2E and Appendix A show the results of common and unique sequence analysis in SNU-719 and AGS cells and their exosomes. The common reading of AGS cells and AGS-exo was about 64.5%, and that of SNU-719 cells and SNU-719-exo was about 71.9%. The common read segment sequence of AGS and SNU-719 cells was 67.1%, while that of AGS-exo and SNU-719-exo was 67.2%. These results suggested that there might be homology between SNU-719 and AGS cells and between their exosomes.

##### Correlation Analysis between SNU-719 and AGS Cells and Exosomes

The correlation coefficient between AGS and AGS-exo ranged from 0.82 to 0.99, and the correlation between SNU-719 and SNU-719-exo ranged from 0.71 to 0.99, showing a high correlation. The correlation coefficients between AGS and SNU-719 ranged from 0.2 to 0.53, those between AGS and SNU-719-exo ranged from 0.2 to 0.3, and those between SNU-719 and AGS-exo ranged from 0.22 to 0.52, both of which showed low correlation. The results showed that the samples and sequencing results were reliable.

##### Differential Expression of miRNA in SNU-719 and AGS Cells and Exosomes

Figure 3 shows the differential expression analysis of human miRNA in SNU-719 and AGS cells and their exosomes. In the AGS cells and AGS-exo, we detected 552 miRNAs, of which 133 miRNAs are statistically significant, 37 miRNAs are highly significant (|log2 (multiple)| >1 and *p* < 0.01), and 96 miRNAs show significant differences ((|log2 (multiple)| >1 and 0.01 ≤ *p* < 0.05). In SNU-719 cells and SNU-719-exo, we detected 551 miRNAs, of which 116 miRNAs are statistically significant, 46 miRNAs show highly significant differences (|log2 (multiple)| > 1 and *p* < 0.01), and 70 miRNAs show significant differences ((|log2 (multiple)| > 1 and 0.01 ≤ *p* < 0.05).

##### Expression Analysis of the Top 15 miRNAs in Cells and Exosomes

Figure 3A shows that the expression levels of hsa-miR-1246, hsa-miR-320a-3p, and hsa-miR-122-5p are relatively high in AGS. The expression levels of hsa-miR-23b-3p, hsa-miR-320a-3p, and hsa-miR-4521 in AGS-exo are also higher. Figure 3B shows that the top three miRNAs highly expressed in SNU-719 are hsa-miR-21-5p, hsa-miR-148a-3p, and hsa-miR-7-5p, and they are also highly expressed in SNU-719-exo.

##### Differential Expression Analysis of EBV-Related miRNAs in SNU-719 Cells and Their Exosomes

Figure 3C shows 43 miRNAs detected in SNU-719 cells and SNU-719-exo, and eleven EBV-miRNAs are statistically significant: five show a highly significant difference (|log2 (multiple)| > 1 and *p* < 0.01), and six show general differences |log2 (multiple)| > 1 and 0.01 ≤ *p* < 0.05). EBV-miRNA was enriched in SNU-719 gastric cancer cells and exosomes, while AGS cells and some cells can also express certain EBV-miRNAs nonspecifically, but there is no significant difference. The expression of EBV-miRNA in SNU-719 cells was the highest in EBV-miR-BART1-5p, EBV-miR-BART22, and EBV-miR-BART16. In SNU-719-exo, EBV-miR-BART1-5p, EBV-miR-BART10-3p, and EBV-miR-BART16 were the highest.

### 3.2. Effects of High and Low Expression of EBV-miR-BART1-3p on Proliferation and Invasion Ability of GC Cells

The cell transfection efficiency of the 5Cy3-labeled inhibitor was observed by inverted fluorescence microscopy. Compared with the control, the results showed that after 24 h, 48 h, 72 h, and 96 h, the 5cy3-labeled inhibitor can persist in intracellular red fluorescence signal, indicating that the inhibitor can continue to enter into cells for 24 h~96 h (Appendix A). The expression of EBV-miR-BART1-3p was determined by qPCR. Figure 4A shows the highest expression level of SNU-719, followed by SNU-719-inhibitor > AGS-OE > AGS-NC, which shows that the inhibitor can reduce the expression level of miR-BART1-3p in SNU-719 cells. These results suggested that SNU-719-inhibitor can be used for the next experiment at 24 h–96 h after transfection.

The CCK8 cell proliferation experiment showed that SNU-719 had the slowest growth rate, followed by SNU-719-inhibitor < AGS-OE < AGS-NC, indicating that EBV-miR-BART1-3p could inhibit cell proliferation (Figure 4B,C).

At 24 h and 48 h after the cell scratch experiment, the healing rate of SNU-719 was lower than that of SNU-719-inhibitor, and the healing rate of AGS-OE was lower than that of AGS-NC, that is, SNU-719 < SNU-719-inhibitor < AGS-OE < AGS-NC, suggesting that EBV-miR-BART1-3p could inhibit cell healing (Figure 4D–F).

The Transwell cell migration assay results showed that the migration rate of SNU-719 was lower than that of SNU-719-inhibitor, and that of AGS-OE was lower than that of AGS-NC, that is, SNU-719 < SNU-719-OE < AGS-NC. It is suggested that EBV-miR-BART1-3p could inhibit cell migration (Figure 4D–I).

Transwell cell invasion assay showed that the invasion rate of AGS-OE and AGS-NC with low expression of miR-BART1-3p was significantly higher than that of SNU-719 and SNU-719-inhibitor with high expression of miR-BART1-3p, that is SNU-719 < SNU-719-inhibitor < AGS-OE < AGS-NC. It is suggested that EBV-miR-BART1-3p could inhibit cell invasion (Figure 4H–J).

### 3.3. Effects of Co-Culture with Exosomes with High and Low Expression of EBV-miR-BART1-3p on Proliferation and Invasion of GC Cells

Exosomes labeled with PKH67 fluorescent dye were co-cultured with cells for 24 h. Laser scanning confocal microscopy revealed green fluorescence signals in the cells, confirming that exosomes could enter them (Figure 5A–D).

In order to determine the optimal working concentration of exosome co-culture of various cells, 0 μg, 5 μg, 10 μg, 15 μg, 20 μg, and 25 μg of exosomes were selected for co-culture and detection of CCK8 proliferation. It was found that the experimental results of SNU-719-exo were statistically different when the working concentration was increased from 10 μg to 15 μg. The experimental results of SNU-719-inhibitor-exo were statistically different when the working concentration was increased from 15 μg to 20 μg. The experimental results of AGS-OE-exo increased from 15 μg to 20 μg and showed statistical difference. The experimental results of exosomes with different concentrations of AGS-NC-exo did not have statistical significance. In order to reduce experimental errors, the working concentration of all exosomes was set at 20 μg based on the above results (Figure 5E).

The relative expression level of miR-BART1-3p in exosomes and in AGS cells after co-culture with exosomes was detected by qPCR, and it was found that SNU-719-exo > SNU-719-inhibitor-exo > AGS-OE-exo > AGS-NC-exo. Similarly, that AGS + SNU-719-exo > AGS + SNU-719-inhibitor-exo, AGS + AGS-OE-exo > AGS + AGS-NC-exo was also found. Combined with the laser scanning confocal results, miR-BART1-3p could be transported into cells through exosomes (Figure 5F–H).

The proliferation experiment of CCK8 showed that the growth rate of AGS + SNU-719-inhibitor-exo was faster than that of AGS + SNU-719-exo (Figure 5I–L), and the growth rate of AGS + AGS-NC-exo was faster than AGS + AGS-OE-exo. These results indicated that EBV-miR-BART1-3p in exosomes could inhibit cell proliferation.

The results of the cell scratch experiment showed that the healing rate of AGS + SNU-719-exo was slower than that of AGS + SNU-719-inhibitor-exo, and the healing rate of AGS + AGS-OE-exo was slower than that of AGS + AGS-NC-exo, that is, miR-BART1-3p in exosomes could inhibit cell healing (Figure 5M–P).

The Transwell cell migration experiment showed that the migration rate of AGS + SNU-719-exo was slower than that of AGS + SNU-719-inhibitor-exo, and the migration rate of AGS + AGS-OE-exo was slower than that of AGS + AGS-NC-exo, that is, miR-BART1-3p in exosomes could inhibit cell migration (Figure 5Q,R,U).

The Transwell cell invasion experiment showed that the invasion rate of AGS + SNU-719-exo was slower than that of AGS + SNU-719-inhibitor-exo, and the invasion rate of AGS + AGS-OE-exo was slower than that of AGS + AGS-NC-exo, that is, EBV-miR-BART1-3p in exosomes could inhibit cell invasion (Figure 5S,T,V).

### 3.4. Changes of Target Genes in Gastric Cancer Cells after Co-Culture with High and Low Expression of EBV-miR-BART1-3p in Exosomes

The target genes ANKRD28, USP37, MACC1, FAM168A, and CPEB3 of miR-BART1-3p were screened by the intersection of miRanda, pita, RNAhybrid, virBase, and TarBase databases (Figure 6).

As shown in Figure 7, CPEB3 was not expressed in all co-cultured cells with exosomes, while ANKRD28, USP37, MACC1, and FAM168A were all expressed to varying degrees. Among them, USP37 and MACC1 were down-regulated after up-regulation of miR-BART1-3p, which may be the key target genes of miR-BART1-3p in regulating the proliferation of GC cells through exosomes.

## 4. Discussion

Etiology, pathology, and molecular characterization of EBVaGC have significant characteristics [19]. At present, there is a lack of qualified animal models of EBV-related carcinogenesis, and EBV-positive tumor cells are used as research models. The most similar EBVaGC cells are the SNU-719 GC cell line, and EBVnGC cells are usually used as control cells.

Epstein–Barr virus miRNAs can produce a state of persistent infection in the human body [20], regulate cell transformation, and participate in the carcinogenic process [21,22,23]. EBVaGC and nasopharyngeal carcinoma can express higher BART clusters [24,25]. miRNAs have also been found to be highly expressed in exosomes, which are involved in intercellular communication [26,27,28,29], inducing cell apoptosis and epithelial-mesenchymal transformation, and promoting the spread and metastasis of GC [30].

Through small RNA sequencing, we found the presence of ncRNAs, including snRNA, snoRNA, tRNA, rRNA, and miRNA, in exosomes isolated and purified from the supernatant of SNU-719 and AGS GC cell cultures. In SNU-719 and AGS GC cells and their secreted exosomes, miRNAs with lengths of 17–27 nt were the most abundant. Annotation analysis showed that sRNAs of SNU-719 and AGS GC cells were mainly located in the exon region of non-coding RNA, and sRNAs of their exosomes were mainly located in the intron region, with highly enriched miRNAs and obvious selective sorting, which was similar to that reported in the literature [31]. We found that AGS-exo highly expressed hsa-miR-23b-3p, hsa-miR-320a-3p, and hsa-miR-4521, and SNU-719-exo highly expressed hsa-miR-21-5p, hsa-miR-148a-3p, and hsa-miR-7-5p.

Analysis of clean reads obtained by small RNA sequencing revealed that all EBV-miRNAs could be found in SNU-719 and its exosomes, among which EBV-miR-BART1-5p, EBV-miR-BART17-3p, and EBV-miR-BART18-5p had the highest expression in SNU-719, and EBV-miR-BART1-5p, EBV-miR-BART18-5p, and EBV-miR-BART17-3p had the highest expression in SNU-719-exo. These selectively packaged and enriched miRNAs are associated with clinicopathological features of EBVaGC.

Shinozaki-Ushiku et al. confirmed the highest expression level of miR-BART1-3p in epithelial tumors [15]. Karimzadeh et al. found that the expression level of EBV-BART1-3p in central nervous system tumor specimens with positive EBV was significantly higher than that in control specimens, and the change in miR-BART1-3p expression was involved in the development of glioma cells [16]. At present, there are few studies on the role of EBV miR-BART1-3p in the development of tumors.

Park et al. transfected miR-BART1-3p into GC cells and found that miR-BART1-3p could induce G0/G1 arrest and inhibit the growth of GC cells. Luciferase assay showed that miR-BART1-3p could directly target the 3′-UTR of E2F3 mRNA. After transfection of miR-BART1-3p, the levels of E2F3mRNA and coding protein were decreased in GC cells, while the expression of E2F3 was enhanced in AGS-EBV cells transfected with miR-BART1-3p inhibitors. These results suggested that miR-BART1-3p may play a role in cell cycle regulation and regulation of miR-17-92 cluster by inhibiting E2F3, affecting cell proliferation and differentiation [17]. Min et al. found that miR-BART1-3p directly targets the 3′-UTR of DAB2, reduces the expression of DAB2 mRNA and protein, increases cell migration, and reduces cell apoptosis. Inhibition of miR-BART1-3p could increase the expression of DAB2, but the effect is the opposite. These results indicated that miR-BART1-3p targeting DAB2 could inhibit cell apoptosis and promote cell migration, playing an important role in the occurrence and development of EBVaGC [18].

By silencing the expression of miR-BART1-3p in EBV-positive gastric cancer cells SNU-719, and comparing with SNU-719, AGS-OE, and AGS-NC, we observed the effects of high or low expression of miR-BART1-3p on the proliferation, invasion, and migration of GC cells. qPCR showed that the expression level of EBV-miR-BART1-3p in SNU-719-inhibitor was lower than that in SNU-719, which confirmed that SNU-719-inhibitor could inhibit the expression of endogenous EBV-miR-BART1-3p in SNU-719 cells. After silencing miR-BART1-3p in SNU-719, the proliferation rate, healing rate, migration, and invasion ability of SNU-719 were significantly improved, suggesting that endogenous miR-BART1-3p could inhibit cell proliferation in EBV-positive gastric cancer cells.

Exosomes were small vesicles actively secreted by cells, with a bilayer membrane structure and a circular, oval, or slightly sunken disk shape [32,33,34]. Exosomes mainly mediate intercellular communication and transfer of bioactive proteins, lipids, nucleic acids (DNA, mRNA, miRNA), etc. [35]. A large number of exosomes with high purity were extracted by ultra-high-speed differential centrifugation of supernatant of cell culture. There were many ways to characterize exosomes [36,37]. By electron microscope observation, we found that there was no significant difference in the morphology of exosomes secreted by EBV-positive and EBV-negative GC cells. The particle sizes of exosomes were in the normal range of 30–150 nm. Exosomes could express proteins with specific phenotypes such as TSG101, Alix, CD63, CD81, and Rab family proteins [38]. By Western blot analysis, we confirmed that two exosomes derived from GC cells expressed CD9 and CD63-specific membrane proteins.

To explore the biological effects of exosomes on tumor cells, most authors add labeled exosomes to the co-culture system to observe the proliferation, invasion, and migration of co-culture cells as well as the changes in the expression of related genes and proteins. At present, PKH67 is among the most commonly used exosome surface fluorescent markers.

Wang et al. found that the transfer of miR-25-3p, miR-130b-3p, and miR-425-5p to macrophages through exosomes and co-culture with CRC can enhance the metastasis ability of CRC [39]. Du et al. co-cultured exosome miR-23a from GC cells with human umbilical vein endothelial cells (HUVECs) and found that VEGF expression was increased, TSP-1 expression was decreased, and PTEN expression was inhibited, suggesting that GC cell-derived exosome miR-23a could promote angiogenesis [40]. Chang et al. transfected MSC cells with lentivirus vectors carrying overexpressed and disrupted sequences of miR-1228 and MMP-14, and then extracted exosomes and co-cultured them with GC cells SGC-7901 and MGC-823 to detect cell proliferation and invasion, apoptosis, and migration. The results showed that miR-1228 can down-regulate the expression of MMP-14 and block the development of gastric cancer [41]. Yoon et al. co-cultured the exosomes of gastric epithelial cells (HFE-145) expressing GKN1 with AGS GC cells, which significantly inhibited the vitality and proliferation, as well as the migration and invasion, of AGS cells [42,43,44].

By extracting exosomes from SNU-719, SNU-719-inhibitor, AGS-OE, immunofluorescence labeling, and co-culture with AGS cells, we observed the uptake of exosomes by AGS cells under laser confocal microscopy. After the detection of miR-BART1-3p in AGS cells co-cultured with exosomes, we found that the expression of miR-BART1-3p in AGS was consistent with that of miR-BART1-3p in the exosomes uptaken, that is, the expression of miR-BART1-3p in AGS co-cultured with exosomes decreased successively as follows: AGS + SNU-719-exo > AGS + SNU-719-inhibitor-exo > AGS, AGS + AGS-OE-exo > AGS + AGS-NC-exo = AGS. The results were similar to those obtained by direct transfection of miR-BART1-3p inhibitors. After co-culture with exosomes secreted by SNU-719-inhibitor, the proliferation rate, healing rate, migration, and invasion ability of AGS cells were significantly improved. These results suggested that miR-BART1-3p can affect the biological function of target cells through the sorting and delivery of exosomes.

Subsequently, through the screening of PITA, miRanda, RNAhybrid, virBase, and DIANA-TarBase databases, we found that FAM168A, MACC1, CPEB3, ANKRD28, and USP37 may be the target genes of miR-BART1-3p. The AGS cells co-cultured with exosomes were detected with qPCR, and it was found that CPEB3 was not expressed, while ANKRD28, USP37, MACC1, and FAM168A were all expressed to varying degrees. The expressions of USP37 and MACC1 were negatively correlated with the expressions of miR-BART1-3p. These results suggest that USP37 and MACC1 may be the key target genes of miR-BART1-3p in regulating the proliferation of GC cells through exosomes.

## 5. Conclusions

EBV-positive gastric cancer cells and their exosomes all expressed human-associated and EBV-virus-associated miRNAs, while EBV-negative gastric cancer cells and their exosomes only expressed human-associated miRNAs. Silencing the expression of miR-BART1-3p could significantly improve the proliferation rate, healing rate, migration, and invasion ability of GC cells. Exosomes could carry miR-BART1-3p into GC cells. By first silencing miR-BART1-3p in exosomes and then co-culturing with tumor cells, its biological effect on GC cells was consistent with that of directly silencing miR-BART1-3p expression in GC cells. It is suggested that GC cells can affect the biological function of adjacent cells through the delivery of miR-BART1-3p by exosomes. Bioinformatics analysis showed that FAM168A, MACC1, CPEB3, ANKRD28, and USP37 were the target genes of miR-BART1-3p. miR-BART1-3p could regulate the expression of USP37 and MACC1 through exosome delivery, which may be a key target gene for miR-BART1-3p to regulate the proliferation of GC cells through exosomes.

## Figures and Tables

**Figure 1 cancers-15-02841-f001:**
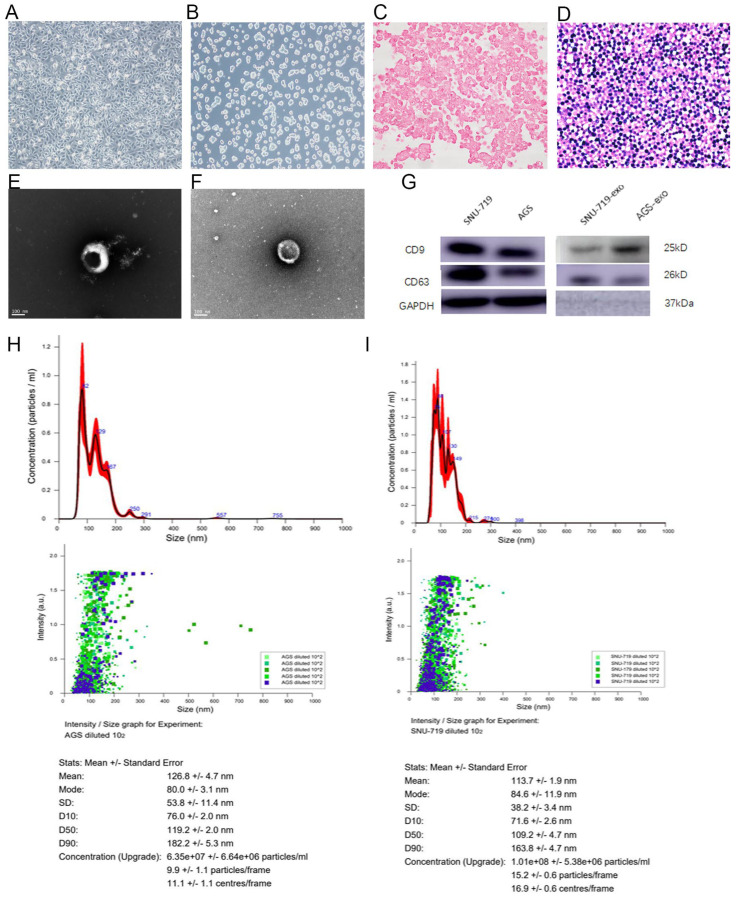
Characterization of morphology, structure, expression, and size of AGS and SUN−719 GC cells and their exosomes. (**A**) Morphology of AGS GC cells; (**B**) morphology of SNU−719 GC cells; (**C**,**D**) EBER in situ hybridization staining showed that AGS was EBVnGC cell line and SNU-719 was EBVaGC cell line; (**E**,**F**) exosome ultrastructure of AGS and SNU−719 cells (98,000×); (**G**) Western blot analysis of specific membrane proteins in SNU−719 and AGS GC cells and their exosomes; (**H**,**I**) NTA results of AGS−exo or SNU−719-exo showed that the exosome particles were evenly distributed with low dispersion and within the normal particle size range.

**Figure 2 cancers-15-02841-f002:**
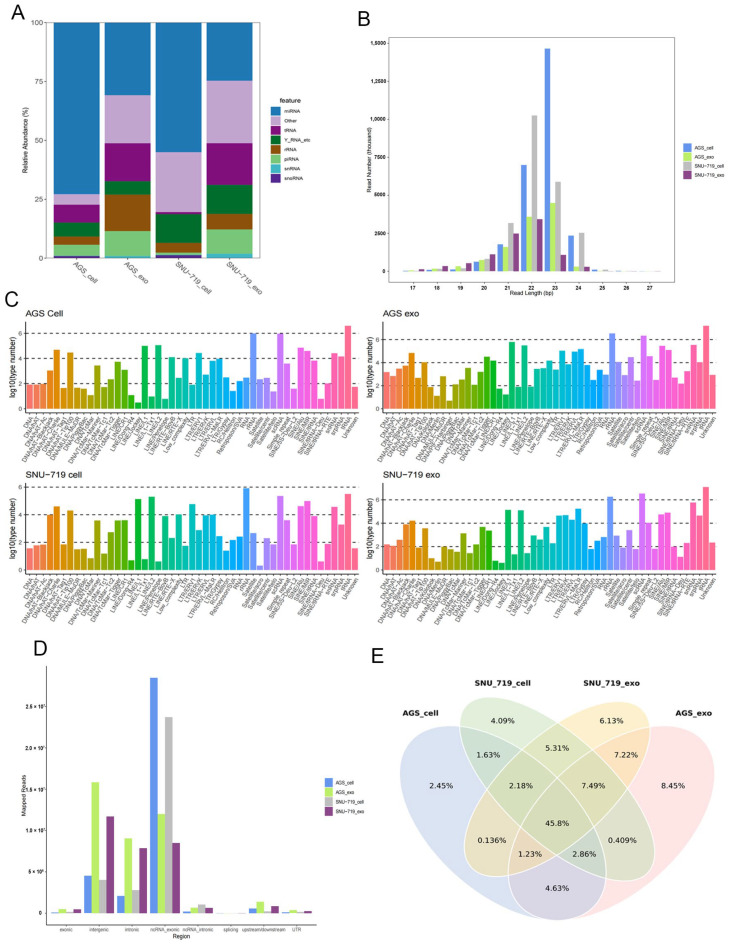
Distribution of small RNAs in cells and exosomes. (**A**) Annotation of ncRNA in SNU-719 and AGS cells and exosomes; (**B**) the small RNA sequence length distribution of SNU-719 and AGS cells and exosomes; (**C**) the sRNA analysis of genomic repeat regions in SNU-719 and AGS cells and exosomes; (**D**) analysis of functional elements in SNU-719, AGS cells, and exosomes; (**E**) Venn diagram of common and unique sequence analysis in SNU-719 and AGS cells and exosomes.

**Figure 3 cancers-15-02841-f003:**
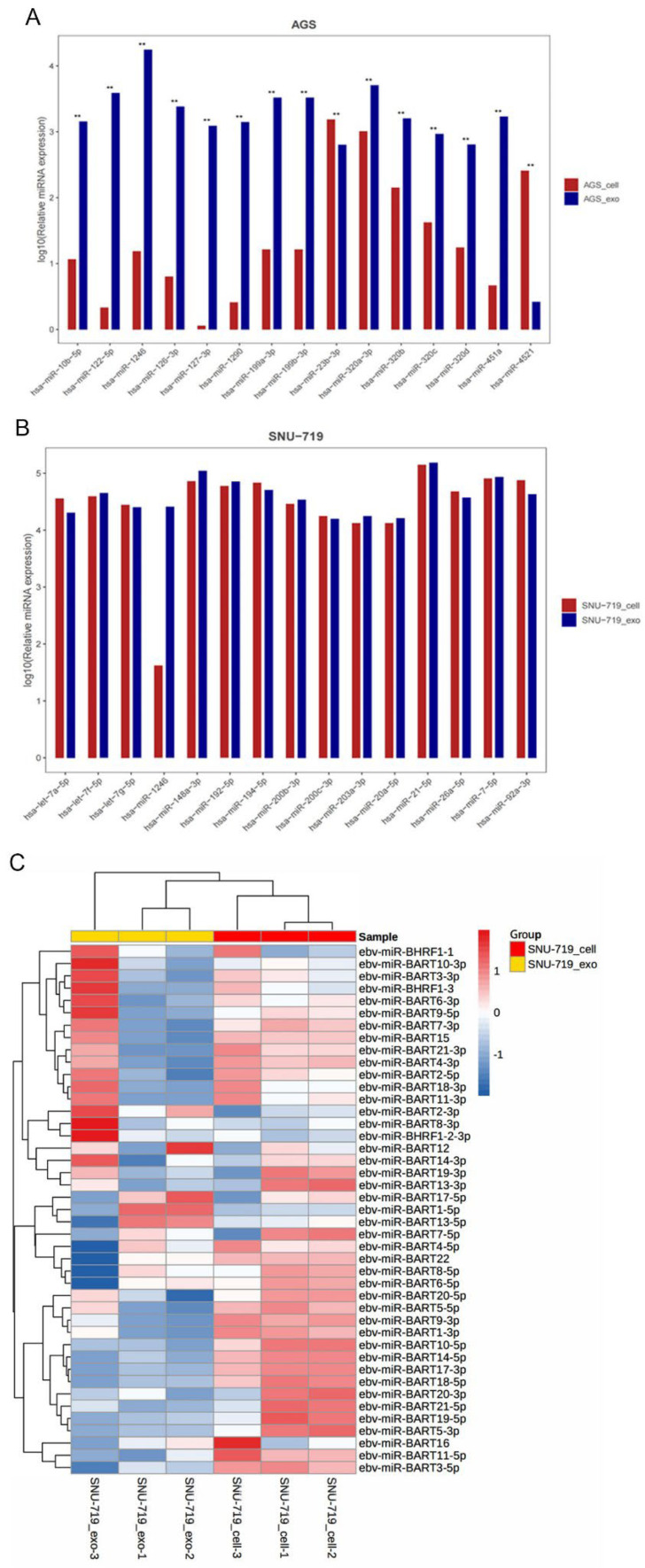
Differential expression of miRNA in SNU−719 and AGS cells and exosomes. (**A**) Expressions of the top 15 miRNAs in AGS cells and exosomes; (**B**) expressions of the top 15 miRNAs in SNU−719 cells and exosomes; (**C**) heat map of the differential expression of EBV-miRNAs in SNU−719 cells and exosomes (Note: ** denotes highly significant difference ((|log2(Fold Change)| >  1 and *p* < 0.01), the same as below.).

**Figure 4 cancers-15-02841-f004:**
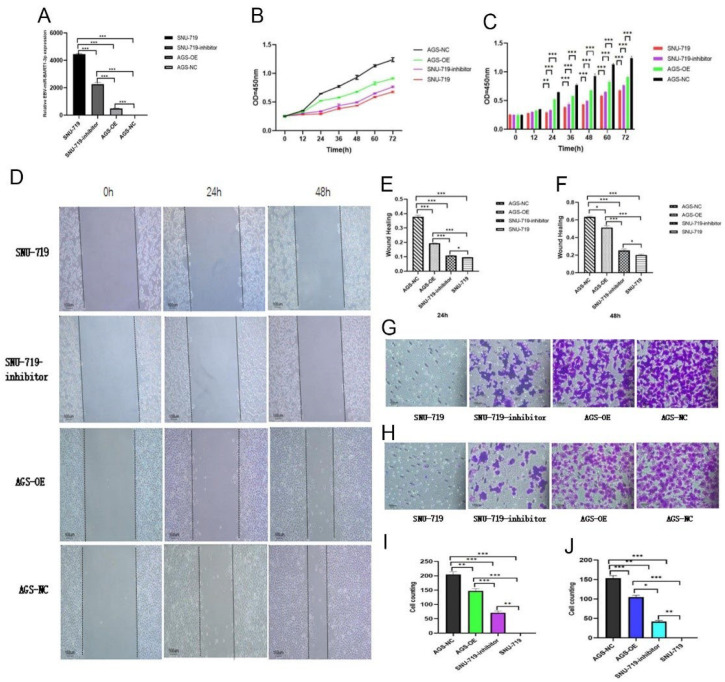
Effects of high and low expression of EBV-miR-BART1-3p on proliferation and invasion ability of GC cells. (**A**) The expression of EBV-miR-BART1-3p was determined by qPCR; (**B**,**C**) the results of the proliferation experiment of CCK8; (**D**–**F**) the results of the cell scratch experiment; (**G**,**I**) the results of the Transwell cell migration experiment; (**H**,**J**) the results of the Transwell cell invasion experiment. (*: *p* < 0.05; **: *p* < 0.01; ***: *p* < 0.001).

**Figure 5 cancers-15-02841-f005:**
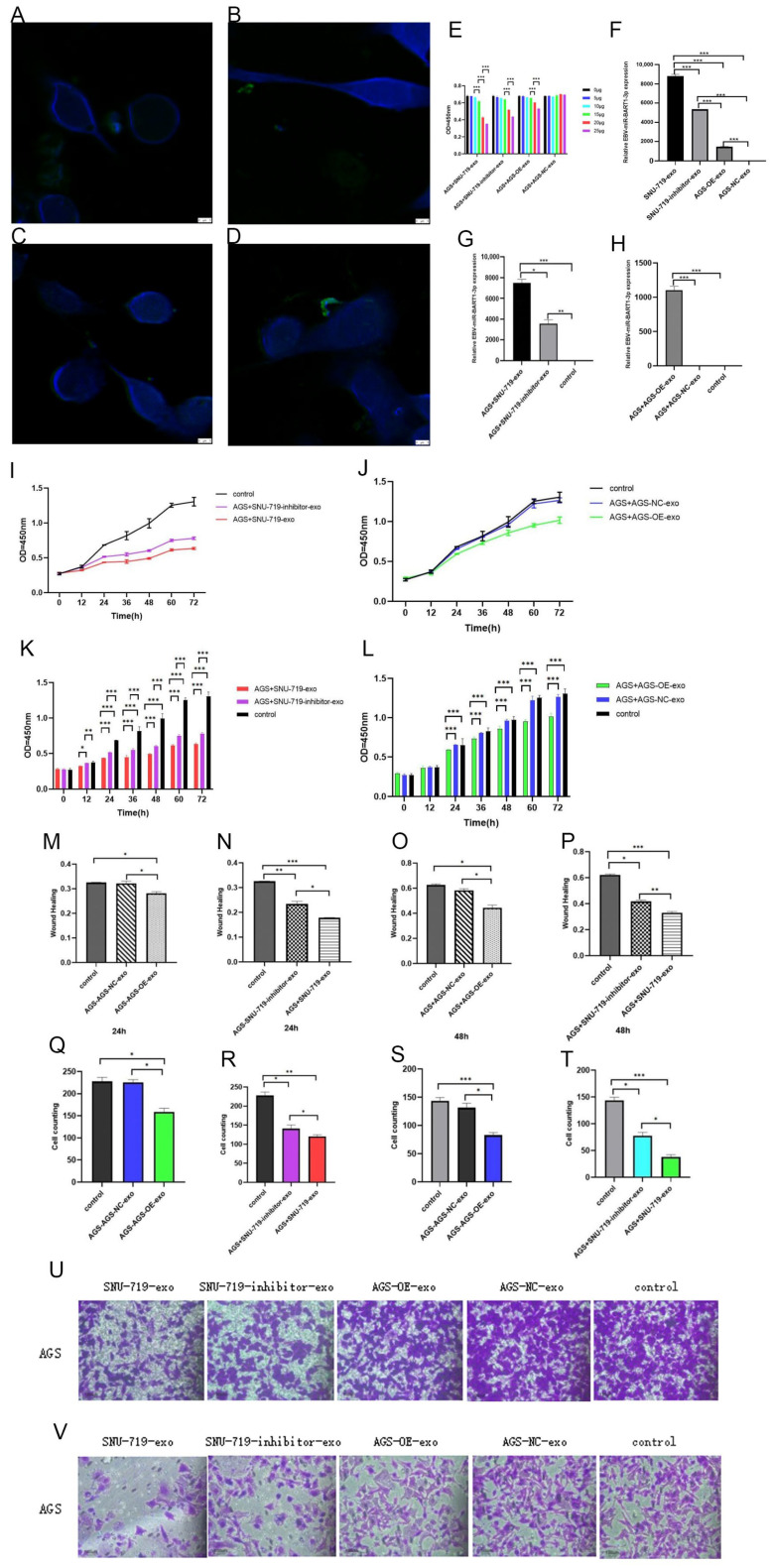
Effects of co-culture with exosomes with high and low expression of EBV-miR-BART1-3p on proliferation and invasion of GC cells. (**A**–**D**) Exosomes labeled with PKH67 fluorescent dye were co-cultured with cells for 24 h. Laser scanning confocal microscopy revealed green fluorescence signals in the cells, confirming that exosomes could enter the cells ((**A**): AGS + SNU-719-exo, (**B**): AGS + SNU-719-inhibitor-exo, (**C**): AGS + AGS-OE-exo, and (**D**): AGS + AGS-NC-exo). (**E**) Determination of optimal working concentration of exosomes; (**F**–**H**) the expression of miR-BART1-3p in AGS cells after 24 h co-culture with exosomes was detected by qPCR; (**I**–**L**) the results of the CCK8 cell proliferation assay; (**M**–**P**) the results of cell scratch assay; (**Q**,**R**,**U**) the results of the Transwell cell migration experiment; (**S**,**T**,**V**) the results of the Transwell cell invasion experiment. (*: *p* < 0.05; **: *p* < 0.01; ***: *p* < 0.001).

**Figure 6 cancers-15-02841-f006:**
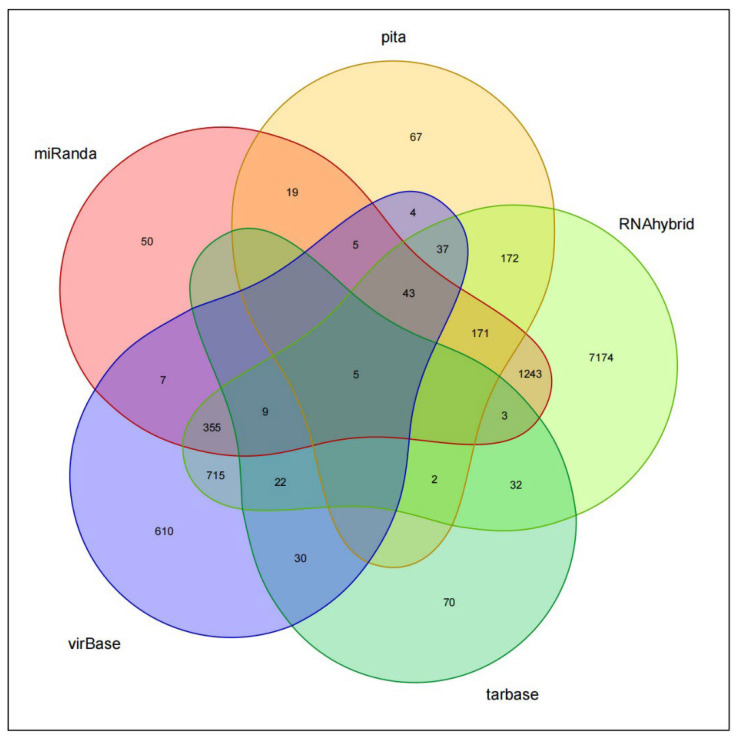
Target gene analysis of miR-BART1-3p.

**Figure 7 cancers-15-02841-f007:**
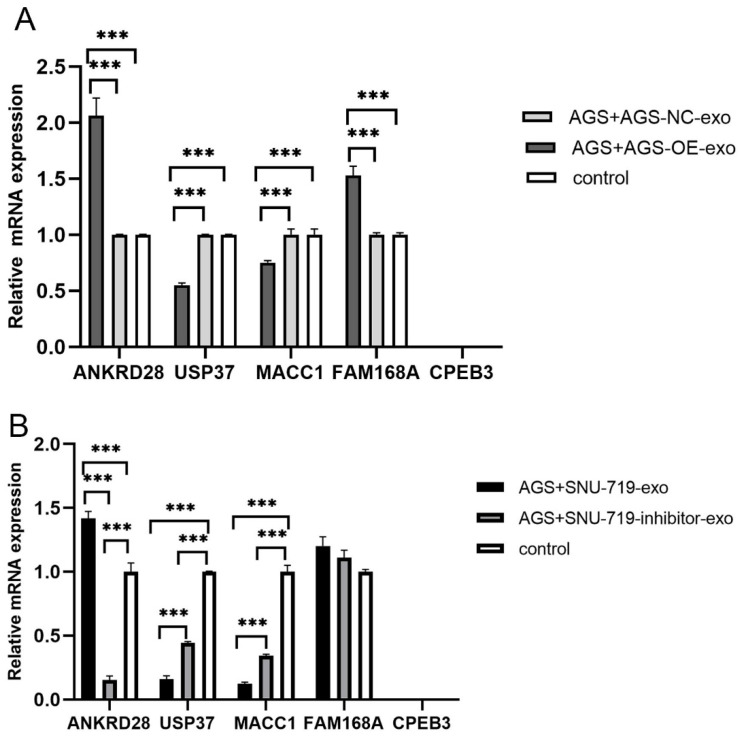
Expression levels of target genes in AGS cells after 24 h co-culture with exosomes (***: *p* < 0.001). (**A**) Expression level of target genes in AGS cells after co-culture with exosomes derived from AGS-NC/OE for 24 h; (**B**) expression level of target genes in AGS cells after co-culture with exosomes derived from SNU-719-exo/inhibitor-exo for 24 h.

## Data Availability

The data presented in this study are openly available in [https://www.ncbi.nlm.nih.gov/geo/query/acc.cgi?acc=GSE227017] (accessed on 13 March 2023).

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
