# Peer review of "Effects of Co-Culture EBV-miR-BART1-3p on Proliferation and Invasion of Gastric Cancer Cells Based on Exosomes"

_cancers, 2023, doi:10.3390/cancers15102841_

Round 1

Reviewer 1 Report

The manuscript needs rewrite in more scientific way, and the language needs  intensively improved by native speaker or professional edit service.

Minor mistakes/typos are everywhere. Please be more serious about the submission and follow the author's guide.

Why there always "briefly as follow" in each paragraph of Method?

Why there are so many multilevel paragraphs in the results?

Letters in Figure and figure legend are impossible to read.

If the format/language in the revised version do not improve by author, I will reject directly without evaluate the scientific soundness.

Author Response

Thank you very much for the revision suggestions. Indeed, although I have written a number of English articles, but for me as a non-native English author, the article is still lack of more scientific. After the revision by the English editor of the magazine, the articles appear much smoother. At the same time, I also carefully corrected the small mistakes in the full text again, removing the expression "briefly as follows" and revising it into a more scientific statement. The English editor also made changes to my multi-level paragraphs and increased the readability of the letters in Figure and legends. Once again, I would like to express my sincere thanks you for your valuable comments. You are my honor.

Reviewer 2 Report

This research article by Lin et al. enumerates a co-culture study to understand the effect of miR-BART1-3p on gastric cancer cell proliferation. The researchers conduct several experiments to prove how this miR candidate affects tumor cell invasion, proliferation etc. The experiments are well thought out and conclusions drawn are supported by their observations. There a few minor aspects that requires more clarification from the authors before it is considered for publication:

1.     In the Materials and methods section: cell culture and reagents- media is wrongly referred to as “PMI”. It’s probably RPMI.

2.     Exosomes were depleted from the FBS used in the experiment. Did the authors perform any experiment to assess the quality of the FBS after exosome depletion? How did they understand that the resultant FBS did not contain exosomes?

3.     Can exosomes derived in their study from culture supernatants be stored long term or do they need to be made fresh for every study?

4.     How did the authors normalize the RNA seq data for cells and exosomes? Did they normalize the number of cells and the number of exosomes used for the sequencing?

5.     Why does the AGS cell miRNAs diff from AGS exo?

Author Response

Thank you very much for the revision suggestions. I tried to answer your question.

1. In the Materials and methods section: cell culture and reagents- media is wrongly referred to as “PMI”. It’s probably RPMI.

Thank you very much for your correction. In the materials and methods section, I did miss a letter "R", causing "RPMI 1640" to be written as "PMI 1640". I have made the correction. My sincere thanks to you.

2. Exosomes were depleted from the FBS used in the experiment. Did the authors perform any experiment to assess the quality of the FBS after exosome depletion? How did they understand that the resultant FBS did not contain exosomes?

Thank you very much for your question. Due to the high cost of commercialized "exosome-free serum", most researchers choose to remove exosomes from serum by means of ultra-fast centrifugation, and several articles have shown that "exosome-free serum" has no effect on cell culture (Such as, Shelke et al. Importance of exosome depletion protocols to eliminate functional and RNA-containing extracellular vesicles from fetal bovine serum. J Extracell Vesicles. 2014;3:24783.) In our previous experiment, we also observed the extracted "exosome free serum" by electron microscopy, and no exosome residue was observed.

3. Can exosomes derived in their study from culture supernatants be stored long term or do they need to be made fresh for every study?

Your question is also the question we had when we were developing the experimental technology route. After reviewing the literature, we found that short-term cryopreservation (within 2 months) did not significantly change the function of exosomes in terms of biological activity for exosome preservation (Zhang et al. Exosome: A Review of Its Classification, Isolation Techniques, Storage, Diagnostic and Targeted Therapy Applications. Int J Nanomedicine. 2020;15:6917-6934.). This result is also confirmed in our experiment.

4. How did the authors normalize the RNA seq data for cells and exosomes? Did they normalize the number of cells and the number of exosomes used for the sequencing?

Your problem is the same problem we had. Later, we carried out normalization analysis of RNA sequencing data under the guidance of technicians. Expression values are normalized by library size, which technically corresponds to miRNA reads per million mapped miRNAs (RPM). The library size is the total number of readings mapped to miRNA precursors. We used miRDeep2 software for normalization analysis.

5. Why does the AGS cell miRNAs diff from AGS exo?

Your question is very pertinent, which is why we are investigating whether the exosome pathway influences the biological behavior of the recipient cells. Exosomes are involved in the delivery of intercellular signals as messengers. Intracellular signaling factors are selectively packaged primarily through intracellular some sorting complex (ESCRT) dependent pathways and then secreted into the extracellular environment and delivered to recipient cells. Therefore, miRNAs in AGS cells and in AGS-exo are inconsistent.

Thank you very much for your guidance and help. We will continue to further study the questions you raised. Thank you very much.

Reviewer 3 Report

The manuscript “Effects of co-culture EBV-miR-BART1-3p on proliferation and invasion of gastric cancer cells based on exosomes“ by Lin et al. analyzed the exosomes of human gastric cancer (GC) cells. The authors detected several human endogenous microRNAs and in the case of the exosomes derived from EBV-positive GC cells  also several ebv-encoded microRNAs by Next-Generation-Sequencing. Further experiments offer novel insights into the role of EBV miRs, especially ebv-miR-BART1-3p for GC related tumor biology including cell migration and cell proliferation.

The first part of the study (exosomes, Next-Gen-Seq) is absolutely thrilling and supports strongly actual research. Very nice!

Unfortunately, the last part completely kills it. Due to wrong or better missing correct controls the entire data of figure 4 and figure 5 can not be reliable interpreted.

The specific inhibitor needs an unspecific control and not the untreated wildtype as control. Also the correct control for the overexpression of the ebv-miR-BART1-3p, which works surprisingly very poorly (see Figue 4A),  should have been the overexpression of a control microRNA or at least an empty vector transfectant and not untransfected cells.

Following major corrections need to be performed:

1.      Focus stronger on the first part of the study. Leave out Figure 4 and Figure 5 or do decrease massively their weighting.

Further minor corrections:

1.      Correct within the abstract: has-miR... into hsa-miR...

2.      Correct line 72: in vitro (italic)

3.      Correct: Leave a space character between number and unit. Sometimes you do and sometimes not.

4.      Correct: Line 125 and line 126 both sentences start with After... (repetition)

5.      EBVnGC (EBV negative GC -OK) and EBVaGC (EBV positive GC-?) What does the a mean?

6.      Figure 2: Interesting information are too small and the uninteresting venn diagramms are much too big. Please correct.

7.      Entire Figure 3 is much too small. Kick out Figure 4 and Figure 5 and increase figure 3.

8.      Please shorten the discussion section.  Focus more on the exosomes and their role for microRNA delivery...etc.

Author Response

Thanks to your guidance, we have corrected the figures and notes in the article to make them more illustrative. In addition, we have read the full text carefully and corrected the errors.

1. Correct within the abstract: has-miR... into hsa-miR...

Thank you for your point. We have checked and modified the miRNAs in the whole paper.

2. Correct line 72: in vitro (italic)

Unfortunately, we changed the in vitro text into italics according to your instruction, but the English editor changed it back. I don't know what I'm gonna do.

3. Correct: Leave a space character between number and unit. Sometimes you do and sometimes not. 

Thank you for your correction. We have carefully checked and corrected the figures and units in the whole paper.

4. Correct: Line 125 and line 126 both sentences start with After... (repetition)

We have also carefully checked and corrected the words in the full text.

5.  EBVnGC (EBV negative GC -OK) and EBVaGC (EBV positive GC-?) What does the a mean?

For EBVnGC and EBVaGC, we added the English full names of Epstein–Barr virus (EBV)-associated gastric cancer and EBV-negative gastric cancer respectively.

6. Figure 2: Interesting information are too small and the uninteresting venn diagramms are much too big. Please correct.

We have corrected Figure 2 to include the boring Venn diagram in the supplementary material.

7. Entire Figure 3 is much too small. Kick out Figure 4 and Figure 5 and increase figure 3. 

We fixed the layout of Figure 3 to make it more prominent.

8. Please shorten the discussion section.  Focus more on the exosomes and their role for microRNA delivery...etc. 

Thank you for your careful correction. This paper aims to study the Effects of co-culture EBV-miR-BART1-3p on proliferation and invasion of gastric cancer cells based on exosomes. If the discussion section is deleted too much, the research results may not be clearly displayed. Therefore, for the discussion part, we only made some modifications, please forgive me. If you feel that we must remove more things, we will follow your instructions.

Once again, I would like to express my sincere thanks you for your valuable comments.

Reviewer 4 Report

I liked the work, it is well written and clear. The results are written well and the discussion considers all the results obtained. In the materials and methods section the specifications of the antibodies used are missing, I would add them for greater clarity.

In the line 110 missing R... (RPMI) I believe.

Author Response

Thank you very much for your guidance and help. We have made detailed revisions according to the opinions of several reviewers. Meanwhile, we have also asked English editors to correct the grammar and content, including mistakes like "In the line 110 missing R" and so on.

Once again, I would like to express my sincere thanks you for your valuable comments.

Round 2

Reviewer 1 Report

Thanks the editing service from MDPI, but I'm afraid the manuscript still need to be rephrased. I will leave the decision of language evaluation for editors.

Experimental design:

Although AGS and SNU-719 are all GC cell lines, the exosomes/cells are not directly comparable in the proliferation/migration assay. The EBV infected AGS is a much better choice, if available. EBV infection will also express other BART miRs and packed into exosomes, as shown in Fig. 3C, which may contribute to any effect in the recipient cells. It will make the results too complicated to make any conclusion.

Why the EBV-miR-BART1-3p was selected in this study? It is not highly expressed in the SNU-719 exosome based on the Fig. 3C, even not in the top 10 list.

Methods:

Still lack some of necessary details to repeat the experiments. For example, the parameter settings of NTA, data processing of small RNA sequencing, PKH67 labeling etc.

What is the reverse transcription method used for miR detection? Small RNAs need a special reaction and primers other than mRNAs.

PBS is not a suitable control in PKH67 labeling experiment. Use dye only control.

Results:

The text in some Figures is impossible to read. Higher resolution pic needed.

CD9 and CD63 blot is not enough to characterize exosome, please see the MISEV guidelines.

How many replicates for miR sequencing? How many wells used in the proliferation/migration assay? Looks too good to be true.

Does any of the target genes related to the function experiments, like proliferation/migration?

Scientific Language and writing:

What is “dense adhesive” in line 184-185 page 4? I GUESS it should be “gel”?? And also “pressure”? “voltage” please.

Some methods need to be combined, like 2.7 and 2.8.

Protein concentration assay is very common in biomedical labs, and do not need to write in details.

High-throughput sequencing is not an accurate expression of what you are doing. Use “small RNA sequencing” instead.

“Gun” in line 240? Really?

Do not use “sucked”, “aspirate” or “pipette” is more like to be used.

Not “Matrigel glue”, only “Matrigel”.

Fonts are not consistent.

Author Response

Dear Sir,

Many thanks to you and the editing service from MDPI for your help. Under your guidance, I read the manuscript carefully again and made some corrections in the manuscript. The introduction was abridged to make it more focused on the purpose of the study.

Due to the lack of eligible animal models of EBV-related carcinogenesis, almost all studies of miR-BART function rely on EBV-infected tumor cell lines. The SUN-719 cell line is currently recognized as one of the most similar EBV-infected cell lines, and recombinant EBV-infected AGS cell lines are also widely used. AGS cells that are not infected with EBV are often used as controll cells. Because recombinant AGS cell lines artificially infected with EBV require a series of complex experimental infection processes and need to be verified, the experiments are relatively tedious. Therefore, we selected gastric cancer cell lines naturally infected with EBV as the experimental object, and the results may be more consistent with the results of naturally infected EBV state.

EBVaGC is a gastric cancer with a background of extensive immune cell infiltration. In our previous study, we found that mRNA expression levels of CXCL10 and CXCL11 in EBVaGCs are higher than those of EBVnGCs, which regulate immune cell migration and differentiation, and may have a potential role in cancer therapy. The expression of CXCL10 and CXCL11 is regulated by EBV-miR-BART1-3p. However, only 5 literatures studied the effects of miR-BART1-3p. Shinozaki-Ushiku et al. found that the expression level of miR-BART1-3p was the highest in epithelial tumors. Similarly, the expression level of miR-BART1-3p in central nervous system tumors was significantly higher than that in normal specimens. Therefore, miR-BART1-3p was selected as the research object.

Thank you for your guidance. We carefully read the experimental method again, and made some key additions to the parameter setting of NTA, data processing of small RNA sequencing, and labeling of PKH67.

The detection of miRNA and the preliminary filtering of small RNAs to obtain clean reading have been supplemented.

PBS is not a suitable control in PKH67 labeling experiment. In most literatures, no control was set for PKH67 labeling. The purpose of our control is simply to show that changes in reaction capacity do not affect the labeling results.

We have changed all the pictures to high resolution ones. Thank you for your guidance.

Exosomes are lipid bilayer structures that can carry proteins such as CD9 and CD63. The detection of CD9 and CD63 is only one means of identifying exosomes. At the same time, transmission electron microscopy is also required to observe whether the extracted products conform to the characteristics of exosomes, namely, membrane structure of lipid bilayer. It is also necessary to use NTA to identify whether the particle size of the extracted product is consistent with exosomes, that is to say, the combination of western blot, electron microscope and NTA can be used to identify exosomes.

For the sequencing of miR, we had 3 duplicate samples. 6 well plates were used for proliferation/migration tests.

We are conducting in-depth studies on these targeted genes proposed in this paper, and the results appear to be related to proliferation/migration. Thank you for your guidance.

“dense adhesive” refers to concentrated adhesive, namely SDS-PAGE concentrated adhesive (5% Acrylamide). In addition, we also have "pressure" changed to "voltage".

2.7 and 2.8 involve the extraction of cellular proteins and exosomal proteins, respectively. The general steps are the same, but there are still small differences. In order to highlight the exosomes that this paper focuses on, it is specially written separately to let readers have a clearer understanding.

We have simplified the expression of the protein concentration assay method. Thank you for your guidance. We have also modified high-throughput sequencing to small RNA sequencing. Change Gun to pipette tip and "Matrigel glue" to "Matrigel". The font of the full text was carefully checked and all the revisions were made.

Thank you again for your guidance.

Sincerely Yours,

Sheng Zhang

Department of Pathology, The First Affiliated Hospital of Fujian Medical University

Reviewer 3 Report

No further comments, the manuscript is now acceptable. 

Author Response

Dear Sir,

Thank you very much for your guidance. We have carefully revised the whole paper according to the comments of all reviewers, including simplifying the introduction, revising the methodological expression, and so on.

Thank you again for your guidance.

Sincerely yours

Sheng Zhang

Department of Pathology, The First Affiliated Hospital of Fujian Medical University
